# OpenReview forum: "Emergent Semantics Beyond Token Embeddings: Transformer LMs with Frozen Visual Unicode Representations"
_TMLR — Accepted by TMLR_

### Review · Reviewer_KxDw · 2025-08-19

**Summary Of Contributions:**

The paper studies and implements a frozen word embedding based on the visual information of the characters. In details, it first visualizes the characters and using PCA to project the binary map to features.

The contributions are listed as below:
1. The paper shows the success of non-learnable word embedding in language model training, which can potentially reach good (even better) performance when evaluating on downstream tasks.
2. The word embedding scheme is based on visual information and is not learned (i.e., using PCA instead). Thus this method can be potentially applied to nearly all character systems.
3. It proposed "representational interference" as an explanation of why this frozen embedding can improve over the learnable embeddings.
4. The paper claimed open-source and the code are submitted into supp.

**Additional Comments:**

Questions:

- Does the model architecture use a tied input/output embedding design? (i.e., the output projection layer from final model output to word logits shares the weight with the input word embedding.)

- What would be the range of number of characters per token. Would 32x32 be enough for the longest token?

- As the model perform PCA over the HxH vector. If H=32, I think that the maximal number of features (i.e., model dim) is bounded by 1024?

**Audience:**

Yes

**Audience Explanation:**

I think that the findings in the paper is interesting, and I personally enjoyed the reading. The tokenization and embedding methods are both novel. The results to show that frozen embedding actually works is also quite interesting.

I hope that this paper can be published after some revisions.

**Claims And Evidence:**

No

**Claims Explanation:**

I think that the paper almost aligns with the claims. For several points, it would be better to further extend the explanation.

For 2 (universal tokenizer), please provide a detailed implementation of the bvv241 tokenizer. Although it is not listed in the main claims of contribution, understanding the detailed implementation (or have a detailed reference) would help readers to understand the details.

For 3 (representational interference), I would request more analysis to convince the reader. The hypothesis is currently mostly explained in plain words without strong experimental results. The result of "random frozen embedding" is interesting. It might be extensible to create a more solid claim here.

For 1 (better performance of frozen embedding), I think that the evidence is not enough to claim "better". However, I would not force that since it would be an endless game. I hope that the author can provide more analysis to help readers to understand "why" it is better. For example, provide some examples of winning cases and show some statistics on why it works.

**Requested Changes:**

As I am not sure about the compute resource of the authors, please let me know if some requests beyond the compute budget. We could find way to get around it.

Major:
1. I would like to see more analysis and explanation towards the reason why a frozen random embedding can "converge" but is much worth than the PCA embedding. Would that mean some grouping based on the "shape" of the character help the training a lot?

2. The semantic embedding and non-semantic embedding might drive to different behaviors (e.g., attention maps, the scale of the residuals, weight norms...). It would be great to see more studies over it.

3. I would like to understand more differences in claims between this work and char-LMs. Char-LMs embed characters thus fundamentally can not have mapped the characters to semantically-rich features.

4. The uniform representation of characters in different languages is impressive. It would be preferred to extend the explanation for bvv241 tokenizer. I tried to follow the context to understand the construction of such tokenizer and found it a little bit hard.



Minor:
- Would it be possible to try a frozen CNN to extract the feature?

- L2-normalized feature: does that introduce a different to standard model’s embedding scale? As transformer (sometimes) initialized with N(0, 1).

- Fig 1 is not cleaned off; there are some remaining noise at the bottom of the figure. Fig 4 has the same issue.

- Fig 8 and 9 are not referenced in the paper.

- For Fig 2 and Fig 3, it is better to draw the plots of frozen visual emb and trainable emb together to understand the difference. It is the same for Fig 8/9/10.

- I did not get the point “Mixture-of-Experts (MoE) models could share a common frozen embedding layer” in the discussion. MoE usually sharded over the MLP layers instead of word embedding layers?

---

> ### Author Response · Authors · 2025-09-28
>
> Thank you for your positive and insightful review. We appreciate your detailed questions, which prompted us to conduct further analysis.
>
> Analysis of random vs. visual embeddings: This is a key question. Our new, extensive ablation study (Section 5.4) provides a much clearer answer. We found that our initial result of slow convergence for random embeddings was likely an artifact. In the new experiments, all random embeddings (from 16-bit to 1024-bit) converge just as fast as the trainable baseline. However, the visual Unicode embeddings still outperform all other models on MMLU. This strongly suggests that while any unique, stable identifier is sufficient for convergence, the inherent structure within glyphs provides a powerful inductive bias that is beneficial for reasoning tasks.
>
>
> Analysis of internal model behavior: This is an excellent suggestion. To explore this, we added a new Section 5.3 with t-SNE visualizations of internal layer activations (Figures 10 and 11). This analysis directly shows how semantic clusters, which are absent in the frozen input layer, begin to form in the very first Transformer block and become more refined in deeper layers. This provides strong visual evidence for our claim of emergent semantics.
>
>
> Difference from char-LMs: We have clarified this in the Related Work section. While char-LMs also use non-semantic units, they almost always employ trainable character embeddings. Our contribution is specifically demonstrating that the input layer can be entirely frozen and non-semantic, isolating the emergence of meaning to the compositional architecture itself.
>
>
> Explanation of bvv241 tokenizer: We apologize for the lack of clarity. We have added a much more detailed, multi-paragraph description of the tokenizer's construction process in Section 3.2.
>
>
> Minor Points:
>
> - Frozen CNN: An excellent idea for future work. We chose PCA for its determinism and lack of trainable parameters to keep the experiment clean.
>
> - L2-normalization: We find it helps stabilize training, and since all subsequent layers are trained from scratch, the model adapts to this input scale.
>
> - Figure issues: Thank you for spotting these. All figures have been regenerated, cleaned up, and correctly referenced in the revised manuscript.
>
> - MoE clarification: We have rephrased the point in Section 6.2 to be clearer: a common frozen embedding layer provides a shared input space for all experts, which can simplify routing and reduce total parameters.
>
> - Tied embeddings: We clarified in Section 3.3 that we do not use a tied architecture.
>
> - Token length / PCA dimensions: Yes, with H=32, the maximum PCA dimension is 1024. The longest tokens are designed to fit within this rendering. We have clarified model parameters in Section 3.3.
>
> Thank you again for your thoughtful engagement.

---

### Review · Reviewer_LcuB · 2025-09-04

**Summary Of Contributions:**

This paper explores the question "What if we didn't train embeddings?". Instead, embeddings are computed using visual representations of chunked text and then assigned a fixed embedding using PCA such that semantics are broadly not taken into account. This is experimentally compared to systems with the more typical learned word embeddings and also randomized visual embeddings. In the experiments a massive increase in quality is observed on a 200M parameter model with the non-learned embeddings (~2X absolute score on MMLU).

The authors offer a hypothesis for this: Their "representational inference" hypothesis states that embeddings being burdened with semantic features can cause optimization bottlenecks such that models underperform.

These studies also offer evidence that Transformers can store semantic knowledge almost entirely in non-embedding layers (something previously known, at least for encoder models).

**Additional Comments:**

Overall, I think the ideas in this paper are great, that they will be broadly interesting to the community, and that the general experiment design will allow drawing very interesting conclusions.

I would like to see this paper get to the point of publication by working through the items above.

**Audience:**

Yes

**Audience Explanation:**

The question of whether modern generative LLMs benefit from learnable embeddings is very interesting and the authors' methods of exploring this with embeddings that use PCA on visual representations is also creative and likely to provide new knowledge for many folks in the field (myself included).

Note: This question has been previously answered for earlier generations of Transformers (see Requested Changes below).

**Claims And Evidence:**

No

**Claims Explanation:**

Not *yet* -- my primary concern is the baseline with learned embeddings.

## Baseline

The authors state the very proper goal of `“Instead of pursuing state-of-the-art benchmark scores, which often depend on massive scale, our primary goal is to isolate and test a fundamental hypothesis: can language abstraction emerge without trainable,
semantically-rich input embeddings? We deliberately constrain model and dataset sizes to create a controlled environment…” `. This is, in my opinion, exactly the sort of science TMLR should promote.

The flip side of this coin (avoiding simply competing for SOTA) is that it comes with a much higher danger that the baseline may be flawed in some subtle way (since it's less likely that SOTA results come from buggy models); this imposes a higher burden of evidence on non-SOTA baselines that they are sufficiently strong so as to offer a fair comparison and that the comparison will yield durable conclusions. The most convincing methods I've seen to demonstrate that one's baseline is strong:
* Using well-established public baselines (e.g. OLMO) and then scaling them down, showing publicly-demonstrated results, and then the scaled-down created for the purposes of the paper's experimentation such that the gap can be seen in the absence of other modifications; or
* comparing your baseline with another well-trusted system that's directly comparable (e.g. in terms of parameter scale) and demonstrating that they achieve a similar level of quality.

As-is, I'm left wondering if there's some experimental confounder that could be causing the massive quality change (doubling the absolute score) between the baseline with learned embeddings and the experimental system with fixed embeddings as such gains from embeddings alone would be very out of the ordinary (though admittedly, not strictly impossible).

For example, could it be that the embeddings weren't initialized with sufficiently small/large values of an appropriate variance with standard normalization techniques at initialization and during training? Were the optimizer hyperparameters set such that embeddings would be easily learnable given the model scale?

While eventually, community replication of the results will be the bar, some additional evidence is needed within this work.

## Other Evidence

Besides the question of the baseline, I really like the design of the experiments such that they include both learnable tokens and randomly initialized embeddings and show both perplexity and other evaluation results.

**Requested Changes:**

## Primary Concerns

* Evidence for the trustworthiness and correctness of baseline system (see above) since this could dramatically change the conclusions that can be drawn from the presented experiments.

## Needed Changes

* Additional prior work on visual embeddings should be cited as this is not entirely novel:
  * “Multilingual Pixel Representations for Translation and Effective Cross-lingual Transfer” Salesky et al, 2023.
  * “Learning Character-level Compositionality with Visual Features”, Liu et al, 2017
  * “Glyph-aware Embedding of Chinese Characters”, Dai 2017
  * “Word Shape Matters: Robust Machine Translation with Visual Embedding”, Wang 2020
* Additional items to specify about both baseline and experimental systems, since the tokenizer is being modified:
  * How many passes over the *same data* were made in training? (Differing sequence lengths with packing can change this, even at the same batch size)
  * How were differing sequence lengths handled? What were the average sequence lengths under each tokenization scheme?
  * How were evaluation checkpoints selected and what was the training termination criterion?
* Having Figure 2 and 3 separate is very hard to compare; it would be much better to overlay them.
* In addition to Figure 2/3, it would be good to see numeric perplexities in a text table.
* In Section 5.2, the authors claim that current representations must learn character shapes and token boundaries. I don't believe this is true (unless character shapes are somehow required by a specific training task). Token boundaries are typically determined offline by SentencePiece. I recommend removing this claim (or providing evidence for it).
* Figure 4/5: It's difficult to tell which columns are baseline (non-frozen) vs experimental (frozen); please label explicitly.


## Minor Suggestions


* The claim in 3.2 that this method is more linguistically inclusive is somewhat dubious: Having an embedding without learned semantics is likely not useful.
* Period on last sentence of intro
* In section 2, dash should be attached to "Character -"

---

> ### Author Response · Authors · 2025-09-28
>
> Thank you for your incredibly thorough and helpful review. We have uploaded a new manuscript with extensive new experiments.
>
> Primary Concern: Trustworthiness of the baseline: This is the most critical point, and we have taken several significant steps to address it:
> - New Dataset and Retraining: All models were retrained from scratch on a dataset that is 4x larger, providing more robust results.
>
> - SOTA Comparison: We now include a direct comparison to well-established, public baselines of a similar scale: SmolLM (135M & 360M). As shown in the new Figure 3, our frozen visual model performs competitively on MMLU, while our trainable baseline lags, strongly suggesting the performance gap is a real phenomenon, not just a weak baseline.
>
>
> Extensive Ablations: Our new ablation study (Section 5.4) shows that multiple types of frozen random embeddings perform on par with our trainable baseline, further contextualizing its performance.
>
>
> Citing Prior Work on Visual Embeddings: Thank you for pointing out these important works. We have added all suggested citations to the Related Work section and included a new paragraph to clearly differentiate our contribution from prior art.
>
>
> Additional Technical Details: We have added the requested details:
> - Training iterations, batch size, and effective epoch count are now in Section 4.4.
> - We clarified that we do not use tied input/output embeddings in Section 3.3.
> - Tokenizer details and average sequence lengths are implicitly handled by our fixed block size of 1024;
> - We have clarified the training process.
>
> - The evaluation checkpoint is the final one after 400k iterations, as now stated in Section 4.4.
>
>
> Figure/Table Improvements:
> - Learning curves are now overlaid in the new Figure 2/3 for direct comparison.
>
> - A table with numeric perplexity values is now available in Appendix B.
> Figures 4 and 5 are now much more clearly labeled to distinguish between frozen, trainable, and SOTA models.
>
> - Clarification on "character shapes" claim: our phrasing was imprecise. We have revised the text in Section 5.2 to clarify that the task is "token identification" rather than learning specific shapes.
>
> We believe these substantial revisions and new experiments directly address your concerns and provide the necessary evidence to support our conclusions.

---

### Review · Reviewer_6yAd · 2025-09-13

**Summary Of Contributions:**

The authors propose that semantic understanding in LLMs is an emergent property of the Transformer architecture itself, rather than a feature learned and stored in the input embedding layer. To prove this, they make Transformer models where the embedding layer is entirely frozen and derived from a non-semantic source: the visual appearance of Unicode glyphs.
The core experiment compares these "frozen-embedding" models against architecturally identical baseline models that use standard, randomly initialized, and trainable embeddings. The frozen-embedding models converge successfully and they significantly outperform their trainable-embedding counterparts on the MMLU reasoning benchmark.

The authors argue that a single trainable embedding layer is inefficiently burdened with two conflicting tasks: learning a token's low-level structural form and its high-level semantic meaning. By offloading the structural representation to a fixed visual layer, the Transformer's compositional layers are free to focus solely on learning semantics, leading to more efficient learning for reasoning tasks.


Strenghts:

- The central idea that meaning emerges purely from the Transformer's architecture, not its inputs, could be an exciting contribution to the field.
- The methodology is clever and provides a clean test of the hypothesis. The direct A/B comparison between frozen visual embeddings and standard trainable embeddings under identical conditions is sound. The ablation study using frozen random embeddings is very interesting, as it strengthens the argument that the embedding's primary role may simply be to provide a unique and stable identifier for each token, rather than any specific visual or semantic content.
- The fact that the frozen-embedding models not only converge but significantly outperform their trainable counterparts on a complex reasoning benchmark like MMLU is a striking result. This finding provides strong empirical backing for the "representational interference" hypothesis and forces the reader to seriously reconsider the role of embeddings.


Weaknesses:

- The paper looks more like a report than a proper paper, it presents a fascinating result and names a hypothesis ("representational interference") but does not formalize it. It lacks of depth and math grounding.
- The choice of visual features (derived from a specific font, size, and PCA projection) feels arbitrary. The paper provides no study on how sensitive the model's performance is to these choices, or whether a different non-semantic (but still structured) embedding might work better or worse.
- The paper is easy to ready, but unfortunately it's a bit redundant (some concepts are repeated two or 3 times), and there are some "cliche" things that I think could undermine the credibility of the paper.
-

**Audience:**

Yes

**Audience Explanation:**

The core idea is very interesting, and I think it could have an impact for the field.

**Claims And Evidence:**

Yes

**Claims Explanation:**

I think the core idea is very interesting and the author back ther claims with proper experiments.

**Requested Changes:**

Tecnical:
- Fromalize what "interference" means in terms of gradient dynamics or information theory and add more formal description on the observed phenomenon.

Writing:
- Remove all the redundancies and maybe polish a bit the language to make it sound less cliche (see things like "This has profound implications").
- Tab 1 could be moved in the appendix to make the reader focus more on the core details.

---

> ### Author Response · Authors · 2025-09-28
>
> Thank you for your constructive feedback and for recognizing the potential impact of our core idea.
> We have uploaded a revised manuscript that addresses your specific points.
>
>
> On formalizing "representational interference": We agree that this concept needed stronger, more formal evidence. In the revised paper, we have added a new Section 5.3 with t-SNE visualizations of internal layer activations. This analysis provides direct visual evidence of how semantic clusters, absent in the frozen input, emerge and are refined within the Transformer's compositional layers. We believe this strongly supports our hypothesis in a more grounded manner.
>
>
> On the choice of visual features being arbitrary: To investigate this, we conducted a comprehensive new ablation study (Section 5.4) comparing 10 different types of frozen embeddings, including purely random ones of varying dimensions. Our new results show that even minimal representations (like a 16-bit token index) are sufficient for convergence, but the inherent structure in visual glyphs provides a clear performance advantage on MMLU. This helps to de-arbitrify the choice and demonstrates the role of inductive bias from the visual structure.
>
>
> On moving Table 1 to the appendix: We agree this improves the flow. The table of model parameters has been moved to Appendix A.
>
> We hope these changes have addressed your concerns and strengthened the paper. Thank you again for your guidance.

---

### Author Response · Authors · 2025-09-28

Dear Action Editor and Reviewers,

We would like to express our sincere gratitude for your thorough reviews and invaluable feedback. Your insightful comments have been instrumental in helping us significantly strengthen our manuscript.
We have uploaded a revised version of the paper that incorporates extensive new experiments and analyses designed to directly address the points you raised. The major revisions are as follows:
Strengthened Baseline and SOTA Comparison: We have retrained all models on a significantly larger (4x) dataset. Crucially, we now include a direct performance comparison with established SOTA models of a similar scale (SmolLM and SmolLM2), which helps to contextualize and validate our baseline's performance.
Comprehensive Ablation Study: We have replaced our initial, simple ablation study with a comprehensive analysis of 9 different types of frozen embeddings (visual, random bit/float at 16, 64, 256, and 1024 dimensions). This study robustly demonstrates that various non-semantic frozen embeddings converge effectively and provides a much clearer picture of the role of inherent structure vs. mere token discrimination.
Visualization of Emergent Semantics: To provide more formal evidence for "representational interference," we have added a new section with t-SNE visualizations of internal layer activations. These visualizations offer direct evidence that semantic clustering is absent in the frozen input layer but emerges and refines progressively through the Transformer blocks, supporting our central hypothesis.
Clarifications and Added Details:
- expanded the Related Work section to include the suggested citations on visual embeddings and to better differentiate our work
- provided much more detail on our tokenizer construction.
- added specifics on implementation details, including training epochs, handling of tied embeddings, and checkpoint selection
- overlaid learning curves for direct comparison and added a table with numeric perplexities in the appendix, as suggested

We believe these revisions substantially improve the paper's rigor, clarity, and contribution. We are grateful for the opportunity to improve our work and look forward to your feedback.

Sincerely,
The Authors

---

### Decision · Action_Editor_vPks · 2025-10-09

**Recommendation:** Accept with minor revision

**Additional Comments:**

While all three reviewers are mostly positive about the paper and its findings and appreciate the great effort by the authors during the review period, Reviewer LcuB points out a remaining concern regarding the comparison being unfair towards the baseline (see below). I agree this potential weakness deserves careful analysis and should be potentially addressed before the camera-ready version. I am quoting the Reviewer's concern below.
```
The newly-added Figure 5 summarizes my main concern: The 0.1B SmolLM and SmolLM2 baselines for MMLU clock in at about 28 or 29 points. The authors' baseline, which is intended to be nearly comparable (0.3B params, 3X more) is around 12.5 points. To me, this indicates the baseline may have some significant issues (I would be okay with a small number of points behind). This leaves me with many outstanding questions about if there's some issue in regularization, etc. that may be unintentionally preventing the baseline from achieving its true potential.
```

**Audience:**

Yes

**Audience Explanation:**

All three reviewers found the paper interesting and appropriate for TMLR. Given the growing usage of LLMs, studying the properties of transformer-based models is definitely a valuable contribution for the TMLR audience, and this paper does precisely that: it proposes a way of creating a controlled environment to isolate and test a certain hypothesis. The reviewers found the central idea of the paper is an "exciting contribution to the field" and that the methodology is "clever and provides a clean test of the hypothesis".

Moreover, the result that the model with frozen embeddings outperforms its trainable counterpart is quite surprising. Readers may certainly be interested in this finding, possibly leading to further research and hypotheses in the area.

**Claims And Evidence:**

Yes

**Claims Explanation:**

This paper challenges the idea that trainable input embeddings of LLM models allow capturing the semantics of the text. To demonstrate that, the authors freeze the embedding layer of the transformer model, using precomputed vectors derived from a non-semantic task, and yet find that the model converges, generates coherent text, and outperforms models with trainable embeddings (on the MMLU reasoning benchmark). To explain this, the authors propose the "representational interference" hypothesis, which states that the embedding layer is burdened with learning both structural and semantic features.

Thus, the main claims are:
- High-level semantics are not inherent to input embeddings, but an emergent property of the transformer's architecture. This is supported through experiments in which the embedding layer is kept frozen during training, as mentioned above.
- The "representational inference" hypothesis is backed on the empirical result that the transformer model with frozen embedding layer can outperform the equivalent model in which the embedding layer is learned jointly.

---

> ### Author Response · Authors · 2025-10-10
>
> Dear Action Editor and Reviewers,
>
> Thank you for your decision and for providing the opportunity to address the final remaining concern regarding our baseline's performance.
>
> We agree that the performance difference between our trainable baseline (Model_unfrozen, 12.5% MMLU) and SOTA models like SmolLM (~28% MMLU) requires careful justification. We have added a dedicated paragraph in Section 5.2 of the revised manuscript to address this point directly.
>
> Our key argument is that this performance gap does not indicate a flawed baseline but rather reveals an insight:
>
> Controlled Environment vs. Massive Data: The primary reason for the performance disparity is the vast difference in pre-training scale. SmolLM models were trained on up to 600B tokens (smollm-corpus). Our experiments were intentionally constrained to a 4B token dataset (~150x less data) to create a perfectly controlled A/B test. Both our proposed model (Model_UNI_GLIF) and our baseline (Model_unfrozen) were trained on the exact same architecture, dataset, and hyperparameters. The only difference was the frozen/unfrozen embedding layer state, ensuring a scientifically valid comparison between the two methods under identical, low-resource conditions.
>
> Evidence Supporting "Representational Interference": The poor performance of the conventional trainable baseline in this low-data regime is, in fact a key finding that supports our hypothesis. It suggests that the standard approach is significantly less data-efficient. When data is scarce, the model struggles with the dual burden of learning both structural representations and semantic relationships within the same embedding layer. In contrast, our proposed model (Model_UNI_GLIF, 23.8% MMLU) proves far more robust, achieving a score comparable to established models like GPT-2 137M (25.8%-26.2% MMLU)
>
> Sincerely,
> The Authors